# Biology of Perseverative Negative Thinking: The Role of Timing and Folate Intake

**DOI:** 10.3390/nu13124396

**Published:** 2021-12-08

**Authors:** Nora Eszlari, Bence Bruncsics, Andras Millinghoffer, Gabor Hullam, Peter Petschner, Xenia Gonda, Gerome Breen, Peter Antal, Gyorgy Bagdy, John Francis William Deakin, Gabriella Juhasz

**Affiliations:** 1Department of Pharmacodynamics, Faculty of Pharmacy, Semmelweis University, Nagyvárad tér 4, H-1089 Budapest, Hungary; petschner.peter@pharma.semmelweis-univ.hu (P.P.); bagdy.gyorgy@pharma.semmelweis-univ.hu (G.B.); juhasz.gabriella@pharma.semmelweis-univ.hu (G.J.); 2NAP-2-SE New Antidepressant Target Research Group, Hungarian Brain Research Program, Semmelweis University, Nagyvárad tér 4, H-1089 Budapest, Hungary; milli@mit.bme.hu (A.M.); gonda.xenia@med.semmelweis-univ.hu (X.G.); 3Department of Measurement and Information Systems, Budapest University of Technology and Economics, Magyar Tudósok krt. 2, H-1521 Budapest, Hungary; bruncsics@mit.bme.hu (B.B.); gabor.hullam@mit.bme.hu (G.H.); antal@mit.bme.hu (P.A.); 4MTA-SE Neuropsychopharmacology and Neurochemistry Research Group, Hungarian Academy of Sciences, Semmelweis University, Nagyvárad tér 4, H-1089 Budapest, Hungary; 5Bioinformatics Center, Institute for Chemical Research, Kyoto University, Gokasho, Uji, Kyoto 611-0011, Japan; 6Department of Psychiatry and Psychotherapy, Semmelweis University, Gyulai Pál utca 2, H-1085 Budapest, Hungary; 7Social, Genetic and Developmental Psychiatry Centre, Institute of Psychiatry, Psychology and Neuroscience, King’s College London, Memory Lane, London SE5 8AF, UK; gerome.breen@kcl.ac.uk; 8UK National Institute for Health Research (NIHR) Maudsley Biomedical Research Centre (BRC), London SE5 8AF, UK; 9Division of Neuroscience and Experimental Psychology, Faculty of Biology, Medicine and Health, The University of Manchester, Oxford Road, Manchester M13 9PL, UK; Bill.Deakin@manchester.ac.uk; 10SE-NAP 2 Genetic Brain Imaging Migraine Research Group, Hungarian Brain Research Program, Semmelweis University, Nagyvárad tér 4, H-1089 Budapest, Hungary

**Keywords:** gene x diet interaction, folate, GWAS, rumination, worry, neuroticism

## Abstract

Past-oriented rumination and future-oriented worry are two aspects of perseverative negative thinking related to the neuroticism endophenotype and associated with depression and anxiety. Our present aim was to investigate the genomic background of these two aspects of perseverative negative thinking within separate groups of individuals with suboptimal versus optimal folate intake. We conducted a genome-wide association study in the UK Biobank database (*n* = 72,621) on the “rumination” and “worry” items of the Eysenck Personality Inventory Neuroticism scale in these separate groups. Optimal folate intake was related to lower worry, but unrelated to rumination. In contrast, genetic associations for worry did not implicate specific biological processes, while past-oriented rumination had a more specific genetic background, emphasizing its endophenotypic nature. Furthermore, biological pathways leading to rumination appeared to differ according to folate intake: purinergic signaling and circadian regulator gene *ARNTL* emerged in the whole sample, blastocyst development, DNA replication, and C-C chemokines in the suboptimal folate group, and prostaglandin response and K^+^ channel subunit gene *KCNH3* in the optimal folate group. Our results point to possible benefits of folate in anxiety disorders, and to the importance of simultaneously taking into account genetic and environmental factors to determine personalized intervention in polygenic and multifactorial disorders.

## 1. Introduction

Neuroticism is a personality trait manifested as a tendency to experience negative, distressing emotions and cognitions [1,2]. The association of neuroticism with depression or anxiety symptoms has been demonstrated to be either fully or partially mediated by rumination, worry, or both [2,3,4,5]. Rumination and worry represent two types of repetitive thought or perseverative cognition. They are negatively valenced, attentive and frequent thoughts about oneself and one’s world which dysfunctionally prolong mental representations of a stressor [6,7,8]. While rumination is considered a past-oriented form of repetitive thought concerning themes of loss and worry is future-oriented negative thinking [7], they share a common latent factor in their variance which predicts future depression and anxiety levels in case of stress [9]. 

Neuroticism and rumination, in addition to their shared phenotypic variance that has important mental health correlates, have been revealed to have a partially overlapping genetic background, which explains a considerable portion of variance in internalizing—but not in externalizing—symptoms [10].

Among genetic associations for rumination [11,12,13], genes of folate metabolism [14] may warrant a special interest. They underpin several biological pathways, properties of which can easily be influenced by diet or supplementation. Low folate levels may be associated with a more severe and endogenous subtype of depression, with marked affective and motivational symptoms and a poorer response to antidepressants [15]. Low folate intake has indeed been associated with an increased risk of later depression [16]. Although results are contradictory [17], folate has been found to be an effective and safe adjuvant antidepressant treatment for major depressive disorder [18,19], especially at a dose of <5 mg/day (or 15 mg/day in case of methylfolate) and as an adjunct to selective serotonin reuptake inhibitor therapy [20]. As a gene-by-environment interaction, folate status may also moderate the effect of folate pathway genes on cognitive flexibility, such that these genetic effects could be detected only in the case of a low erythrocyte folate level [21].

Although a genome-wide association study (GWAS) was recently published on rumination and its two subtypes [11], the role of folate intake has never been investigated within this framework. Our present aim was a GWAS on the two perseverative negative thinking items of neuroticism scale, within separate groups of individuals with suboptimal versus optimal folate intake, in the UK Biobank database (http://biobank.ctsu.ox.ac.uk/, accessed on 27 November 2019). We expected distinct genetic associations in different folate intake groups. We also aimed to test the explanatory value of top risk variants on neuroticism score, rumination score and scores of the two subtypes of rumination in a separate database. We expected shared genetic factors between the two perseverative negative thinking items and the investigated comprehensive personality traits. Participants’ current depressive symptoms and lifetime depression status were controlled for in all analyses, to uncover the potential transdiagnostic relevance of the endophenotype of perseverative negative thinking beyond its importance in depression.

## 2. Materials and Methods

### 2.1. Participants

Invitation and recruitment in the UK Biobank study were based on NHS patient registers of people aged 40–69 years [22], with an ethical approval from the National Research Ethics Service Committee North West–Haydock [23].

In the NewMood study, participants aged 18–60 years were recruited through advertisements, general practices and a website, with ethical approvals from the North Manchester Local Research Ethics Committee, Manchester, United Kingdom, and the Scientific and Research Ethics Committee of the Medical Research Council, Budapest, Hungary [11].

All participants in both databases provided written informed consent. All procedures were carried out in accordance with the Declaration of Helsinki.

Our study was based on UK Biobank Application Number 1602, which focused on the Oxford WEbQ dietary questionnaire data, completed by a subset of UK Biobank participants. In our present analyses, we included white British UK Biobank participants who had passed genomic quality control (QC) and had non-missing values on all of the variables of interest (both perseverative negative thinking items, folate intake, sex, age and both depression measures, as detailed below). White British ancestry was defined by a self-report and genetic ancestry in data-field 22006, and we decided to analyze only this population because it is the one that constitutes the vast majority of the UK Biobank database [24]. All of these restrictions yielded a total of 72,621 participants in our analyses. Of these, 10,638 subjects belonged to the suboptimal folate intake group and 61,983 belonged to the optimal folate intake group.

Within our NewMood database, we restricted our present analyses to a set of 1746 subjects that were very similar to those in our previous rumination GWAS paper [11], specifically, white Europeans from Manchester, United Kingdom and Budapest, Hungary, who did not overlap UK Biobank participants, passed genomic QC and provided information on sex, age, both depression phenotypes (detailed below), as well as on rumination and neuroticism.

### 2.2. Phenotypes

In UK Biobank, the two phenotypes on perseverative negative thinking were two items from the neuroticism scale of the Eysenck Personality Inventory, assessed at baseline [22,23,25]: “Do you worry too long after an embarrassing experience?” (data field 2000; rumination item, because of its orientation towards the past) and “Are you a worrier?” (data field 1980; worry item, because its content is restricted to this trait). Both items are used as dichotomous variables: their values can be either yes or no.

In UK Biobank, current depression level was determined by the sum of four item scores [26] detailed in Appendix A, each one measured at a four-point Likert scale at the baseline time point. Lifetime depression status was assessed also at baseline, by a verbal interview (data field 20002).

Folate intake in UK Biobank was estimated via food and beverage consumption the day before, excluding any supplements (data field 100014). It was based on Oxford WEbQ, a 24 h dietary recall questionnaire, which was validated against an interviewer-administered 24 h dietary recall [27]. To control for seasonal variability of dietary intake, more time points of assessment were used, with a maximum of five. 41.82% of participants provided dietary data at only one time point, 20.28% had two, 18.29% had three, 14.45% had four, and 5.15% had five instances of assessment. 59.78% of all participants provided dietary data at the baseline time point, others only at online follow-up instance(s). Values over multiple instances were averaged for each subject, and nutrient calculation has been detailed elsewhere [28]. Cut-off point of a healthy level of folate intake was set to 200 µg per day. This value was based on the recommendation of the British government, as detailed in Appendix A.

In our NewMood database, folate intake was not measured. However, we had measures on comprehensive scales of personality traits. Neuroticism was measured by the 44-item Big Five Inventory (BFI-44) [29]. Rumination and its two subtypes were assessed by the 10-item Ruminative Response Scale (RRS) [30]. Five items belong to the brooding subscale, which denotes a “moody pondering”, passive comparisons of the person’s current situation with unachieved standards [30]. The other five items of RRS belong to the reflection subscale, denoting a purposeful turning inward with the aim of problem solving and alleviating depression [30].

In NewMood, current depression level was addressed by the sum of depression item scores and additional item scores of Brief Symptom Inventory (BSI) [31]. Lifetime depression status was measured by a self-reported question within the background questionnaire and was validated with diagnostic interview within a subsample [32].

In order to handle item-level missingness properly for neuroticism, rumination, brooding, reflection, and BSI depression in NewMood, and current depression score in UK Biobank, the sums of item scores were divided by the number of responded items.

### 2.3. Genotyping, Imputation and Genomic Quality Control

In UK Biobank, we selected participants with a genetically defined white British ancestry subset (data field 22006), without putative sex chromosome aneuploidy (data field 22019), and, as part of a further QC process, those involved in the maximal set of unrelated individuals (data field 22020) [24].

In both UK Biobank release v3 and our NewMood database, genetic variants were restricted to biallelic single-nucleotide polymorphisms (SNPs), analyzing directly genotyped and imputed variants as well. QC steps are detailed in Appendix A. 

### 2.4. Analyses

Descriptive statistical analyses, as well as calculation of standardized residuals for polygenic risk score (PRS) testing were performed with SPSS 25. LD Score regression [33] was used to calculate the SNP heritability of rumination and worry; namely, the phenotypic variance explained by the whole set of SNPs in each of the three groups of UK Biobank participants (whole study sample, suboptimal folate intake group, optimal folate intake group). In each group, the top ten principal components of the genome were calculated with the approximative method [24,34] implemented in Plink2 (www.cog-genomics.org/plink/2.0/, accessed on 26 March 2020) [35], after respective linkage disequilibrium (LD) pruning in that group. Plink v1.9 (www.cog-genomics.org/plink/1.9/ (accessed on 26 June 2020)) [35] was used to explore identity-by-descent parameter to detect potential overlap between UK Biobank and NewMood participants.

For SNP-level analyses, logistic regression models were run in Plink v1.9, for each of the rumination or worry items as outcome, in all three groups according to folate intake. We applied the strictest method to correct for multiple testing: the Bonferroni correction. The number of tests entailed a *p* ≤ 1.38 × 10^−9^ significance threshold (detailed in Appendix A). 

To go further with these SNP-level results, FUMA v1.3.6 [36] was used. Within FUMA, MAGMA v1.07 [37] was used for gene-based and gene set-based analyses, and for gene property analysis that assessed tissue specificity. Gene-based testing assigned SNPs to protein-coding genes based on position, with gene boundaries extended by 10,000 base pairs, and entailed a *p* ≤ 4.36 × 10^−7^ as a significance threshold by the strictest Bonferroni method. To further analyze the resulting gene *p*-values, gene set testing used 15,496 gene sets from MsigDB v7.0: 5500 C2 curated gene sets and 9996 C5 GO terms, yielding a *p* ≤ 5.38 × 10^−7^ significance threshold, again by the strictest Bonferroni method. To explore associations between gene *p*-values and tissue specificity of genes, expression databases used for testing included the 30 general tissue types of GTEx v8 [38] and the 11 general developmental stages of BrainSpan’s developmental brain samples [39], entailing a 2.03 × 10^−4^
*p*-value threshold with the strictest Bonferroni method.

In additional FUMA analyses, to overcome the limitations of using only position-based assignment and protein-coding genes within MAGMA, top SNPs of SNP-level tests were mapped to genes based not only on position, but also on functional annotations: expression quantitative trait loci (eQTL) and 3D chromatin interaction. These mapped genes were then further analyzed with hypergeometric tests if overrepresented in any pre-defined gene set of the above detailed MsigDB gene set collections. As a built-in function of FUMA “GENE2FUNC” analysis, a Benjamini–Hochberg false discovery rate (FDR); *p* < 0.05 significance criterion was applied within each subcategory of gene sets.

Further details of FUMA analyses are provided in Appendix A.

PRSice-2 [40] was used for PRS analyses. Explanatory value of PRS composed of each of UK Biobank’s six SNP-level logistic regression analyses was tested for standardized residuals of NewMood’s neuroticism, rumination, brooding and reflection scores. To control for false positive results due to multiple testing, the built-in permutation function of PRSice-2 was used to run 10,000 permutations for each of these 24 models, and models with a resulting empirical *p* ≤ 0.05 were considered significant.

PRS calculation, using the same procedure, was also applied to test the explanatory value of top SNPs within the suboptimal folate intake group for the same perseverative negative thinking item within optimal folate intake group, and vice versa.

A more detailed description on PRS calculation is provided in Appendix A.

Figures on results were exported from FUMA or PRSice-2, and labels were added to them in Microsoft Word and PowerPoint.

## 3. Results

### 3.1. Descriptive Statistics and Single-Nucleotide Polymorphism (SNP) Heritability

In the UK Biobank, folate intake was inversely related to worry, but unrelated to rumination. The NewMood sample was younger, more depressed and more predominated by females than the UK Biobank sample. Descriptive statistics and relationships between phenotypes are further detailed in Appendix A of Appendix A.

Table 1 depicts proportions of phenotypic variance explained by the whole set of SNPs.

### 3.2. SNP-Based Results

Appendix A of Appendix A shows that no SNP survived genome-level Bonferroni correction for multiple testing. QQ plots and lambda values for each analysis are provided in Appendix A of Appendix A.

### 3.3. MAGMA’s Gene-Based Results

Only two genes survived correction for the six tests: *ARNTL*, for rumination in the whole sample (Figure 1a), and *KCNH3*, also for rumination, in the optimal folate intake group (Figure 1b). Manhattan and QQ plots of gene-based *p*-values for the two items in the three groups, are shown in Appendix A of Appendix A. Gene-based results are detailed fully in Sheet 1 of Appendix A.

### 3.4. MAGMA’s Gene Set-Based Results

No gene set survived multiple testing correction. Sheet 2 of Appendix A detail full gene set-based results for the six analyses. Among folate-related pathways, only the two gene sets related to transmembrane folate transport emerged in the top few ones, and only for rumination in the whole sample. Specifically, these ranked at positions 24 and 34 (Sheet 2 of Appendix A).

Homocysteine metabolism is closely linked to folate metabolism and it emerged in top ten pathways—but only of rumination and only in the suboptimal folate group, as Table 2 points out, along with ranks of these top ten pathways in the other five groups.

### 3.5. MAGMA’s Tissue-Specific Upregulation of Genes Relevant in Rumination or Worry

Gene property analysis for tissue-specific upregulation uncovered no significant results. However, at a nominally significant level, Figure 2 depicts results indicating that genes that showed strong association with rumination in the suboptimal folate group were highly expressed during late-mid and late prenatal brain development. Moreover, also at a nominally significant level, genes with strong association with worry in the whole study sample were highly expressed in the pancreas (Appendix A of Appendix A). Results are fully detailed in Appendix A.

### 3.6. Genes Mapped Both by Position and Functional Annotations, and Their Enrichment in MsigDB C2 and C5 Gene Sets

For each of the six analyses, Sheets 3 and 4 of Appendix A (and Appendix A) detail genes mapped to top SNPs, based on expression quantitative trait loci (eQTL) and 3D chromatin interaction databases, respectively. These full results are sorted according to tissue or cell type and gene names within each tissue or cell type.

Enrichment tests of mapped genes revealed a possible significance of purinergic signaling in rumination in the whole study sample (Appendix A of Appendix A), although the involved genes (*P2RY1*, *P2RY12*, *P2RY13*, *P2RY14*, *GPR87* and *GPR171*) all reside in the q25.1–2 regions of chromosome 3 (https://genome.ucsc.edu/ (accessed on 14 May 2020)). Similarly, for rumination in the suboptimal folate group, diverse immune system gene sets, ERK cascade, regulation of DNA replication, and blastocyst development might be implicated (Appendix A of Appendix A), but most of these genes (*CCL2*, *CCL7*, *CCL11*, *CCL8*, *CCL13*, *CCL1*, *MMP28*, *ZNF830*, *LIG3*, *SLFN11*, *NLE1*, *HNF1B, RFFL*) are located in the same region of chromosome 17q12. For rumination in the optimal folate group, some general pathways emerged, such as responses to fatty acids and prostaglandin (Appendix A of Appendix A) involving *PTGDR*, *PTGER4*, *GNG2*, *APOB* and *CREB1*, among others.

For worry, significantly enriched gene sets are detailed in Appendix A of Appendix A. The majority of these very few hits were related to cancer diseases.

### 3.7. Explanatory Value of UK Biobank’s Risk SNPs in NewMood’s Phenotypes

Polygenic risk scores (PRS) that had been calculated based on UK Biobank’s risk SNPs for rumination and worry items in the three (whole sample, suboptimal and optimal folate intake) groups, were tested for neuroticism, rumination, brooding and reflection scores of the NewMood sample. Sheets 5–8 of Appendix A show that from these 24 analyses, only one result survived permutation: the most significant (with a *p*-value inclusion threshold of 5.01 × 10^−5^) variants (*n* = 23) that denoted a risk for rumination item in the optimal folate group significantly explained 0.56% of variance in the rumination scale (Appendix A).

### 3.8. Potential Overlap in Risk SNPs of “Rumination” or “Worry” between Suboptimal and Optimal Folate Intake Groups

PRS based on the suboptimal folate intake group explained 0.054% of rumination and 0.051% of worry within the optimal folate intake group. In contrast, PRS based on the optimal folate intake group explained 0.3% of rumination and 0.35% of worry within the suboptimal folate intake group (Sheet 9 of Appendix A). All these results are significant at a permuted *p* = 1 × 10^−4^ level.

## 4. Discussion

Perseverative negative thinking is a complex cognitive process leading to several somatic and mental disorders. Our study demonstrated that the past-focused rumination component of perseverative negative thinking has a more specific genetic background compared to general worry, which seems more heterogeneous. Namely, our data, in line with previous findings, suggested that the *ARNTL* gene and purinergic genetic pathway contributed to rumination in the whole sample. However, when we focused on the subgroup of suboptimal folate intake, well-known folate-dependent pathways emerged in the background of rumination, such as regulation of DNA replication, blastocyst development, homocysteine metabolism, and diverse immune responses including chemotaxis, ERK cascade, interferon-, interleukin 1- and tumor necrosis factor-response. Furthermore, genes that were strongly associated to rumination in the suboptimal folate intake subgroup are involved in brain development during late-mid and late prenatal period, emphasizing the role of these genes not only in brain development but in adult cognitive processes. Polygenic genetic compositions of rumination within different folate intake groups explained much less variance from each other than the total set of SNPs from rumination within each group. These observations suggested that lack of adequate folate intake can divert biological processes in the background of rumination.

### 4.1. Time Perspective and Event-Specificity of Perseverative Negative Thinking

In our study, significant and plausible results emerged only for rumination, i.e., no such results were found for worry. Nagel et al. [23] conducted analyses on the same neuroticism items within the same UK Biobank database as our present work, and with a considerable overlap in analysis methods. Results for rumination seemed more consistent across the two studies of different sample sizes and different covariates than results for worry (discussed in detail in Appendix A).

Content of the two items (“Do you worry too long after an embarrassing experience?” and “Are you a worrier?”) differ from each other only in that the former has a focus on specifically defined past events, while the latter has no specific time or event focus. The “Are you a worrier?” item could itself possibly be further decomposed into distinct elements with divergent underlying cognitive processes, thus somewhat impeding the delineation of a compact, well-defined genetic background. The rumination item, in contrast, represents a well-defined target and mode of perseverative negative thinking, and may involve specific deficits of cognitive control and memory—namely, difficulties in discarding no longer relevant negative information from working memory [13,41]. Because of these specific cognitive underpinnings, in addition to more robust results across studies, it may be a better endophenotype [42] than the more general and wide concept of worry.

A further difficulty in the detection of replicable genetic underpinnings of worry may be that worry or its distinct components might be more dependent on interactions between different genetic variants with each other or with external factors. 

Nevertheless, folate intake itself was inversely associated only with the more general item of worry, but not with the more specific form of perseverative negative thinking, rumination. Previous studies have yielded contradictory results on the associations between folate levels and anxiety disorders. While an early study found decreased serum folate level in adult patients with obsessive-compulsive disorder (OCD) compared to controls [43], no later studies could replicate this finding in pediatric OCD [44], in generalized anxiety disorder (GAD) [45], or in higher anxiety score within a cohort study [46]. On the other hand, the same studies found higher homocysteine levels in patient groups of adult OCD [43], pediatric OCD [44], and GAD [45]—but again, no association with anxiety level within a cohort study [46]. Similar to results on folate levels in patient groups, results have also been contradictory on the beneficial effects of adjuvant folates in treatment. While a retrospective study suggested the benefits of l-methylfolate calcium in multimorbid pediatric patients [47], a double-blind controlled study found no differences between placebo group and folic acid group in OCD patients treated with fluoxetine [48]. Further studies are needed to clarify the associations between folate and the emergence (as well as treatment) of different kinds of anxiety disorders, and our results within a large population study may fuel these investigations.

### 4.2. Genetic Determinants of Past-Focused Perseverative Negative Thinking

Nagel et al.’s [23] rumination results were in line with our whole study sample’s results. In our study, despite the lack of association between folate intake and rumination, nominal significance of gene sets related to transmembrane folate transport for rumination in the whole sample suggested the importance of folate metabolism in this form of perseverative negative thinking. Moreover, we could reveal different biological mechanisms in the background of rumination in different groups stratified according to folate intake. Herein, we proposed a summary model to depict the complexity of the pathomechanisms behind rumination, detailing its currently identified aspects and integrating our results with the results of former studies (Figure 3). Nevertheless, future studies are also needed to uncover causal directions in these gene-by-diet interactions in the background of rumination.

We postulated that there may be some general, umbrella-like, multifaceted biological mechanisms in the background of rumination, which affect several distinct, narrower processes and pathways. As such, *ARNTL* (or *BMAL1*) gene emerged for rumination in the whole sample. This gene is a key positive regulator of the circadian clock, showing a circadian oscillation in transcription and translation, and this rhythm then regulates the expression of more than 10% of the transcriptome [49]. *ARNTL* is a shared gene between rumination and several neuropsychiatric disorders [49,50,51,52,53] and obesity [54], suggesting this past-focused form of perseverative negative thinking as a potential transdiagnostic endophenotype. Another umbrella-like mechanism behind rumination, detected only in the whole sample, may be the importance of purinergic signaling. Genes of P2Y purinergic receptors, implicated in our results and in animal studies of perseverative cognition [55,56], may exert their effects on rumination through divergent biological pathways [11,12,32,57,58,59,60,61]. Purinergic effects on each of these narrower pathways are hypothesized to point to the same direction regarding the generation of rumination. These effects add up and convey a significance to the general, umbrella-like mechanism. However, external factors such as folate intake may modulate each of these distinct, narrow pathways in a different magnitude or manner [62,63,64]. Consequently, in determining the level of rumination in cases of different folate statuses, some of these pathways will become less important, while others will become more so.

### 4.3. Genetic Determinants of Past-Focused Perseverative Negative Thinking in Case of Suboptimal Folate Intake

In the suboptimal folate intake group, the genetics of rumination may be best determined by the specific pathways and processes most sensitive to folate deficiency. Indeed, DNA synthesis and repair [65], early prenatal development [66], prenatal brain development [67] and levels of C-C chemokines [68] and homocysteine [69] have been demonstrated to be influenced by folate levels, even in interaction with genetics [63,70,71], and have shown associations with neuropsychiatric phenotypes [63,72,73], as well. Adverse effects of genetic, intrauterine or immunometabolic factors on these phenotypes may be compensated by sufficient folate intake, although our results suggested that developmental timing is crucial in folate’s compensatory impact in case of these biological pathways and rumination, as adult folate intake has not showed association with rumination. In line with our results, cohort studies suggested an association between conception during a severe famine and increased prevalence of neurodevelopmental disorders, such as neural tube defects, as well as schizophrenia and schizoid personality [74]. Somewhat contrary to our results, independent additive effects of genetics and prenatal nutrition, not gene-by-nutrition interaction, on childhood processing speed have been suggested by a randomized, placebo-controlled, double-blind study [75]. 

Concurrent folate intake was not associated with an event-specific past-focused form of adults’ perseverative negative thinking in our study, perhaps because the effect of folate intake on rumination is not linear and may depend on some genetic vulnerabilities or other factors. It is important to highlight however, that folate has been suggested to impact the nervous system at all ages [15]. Higher plasma folate level has been associated with better global cognitive functioning and faster psychomotor speed as well as a reduced risk of severe white matter lesions in non-demented elderly participants [76]. Although a study did not find any associations between folate status and cognitive dysfunction or brain atrophy in Alzheimer’s patients or controls [77], a randomized, double-blind study revealed that treatment with B vitamins slowed brain atrophy [78]. Hemodynamic activity in the brain can indeed serve as an intermediate phenotype between B vitamins and cognitive functioning [79].

### 4.4. Genetic Determinants of Past-Focused Perseverative Negative Thinking in Case of Optimal Folate Intake

In cases of optimal folate intake, genetic vulnerabilities within numerous specific pathways may be compensated, and only narrow specific processes emerge in genetics. Hippocampal neuronal excitability [80] may be such a mechanism, which was reflected in voltage-gated K^+^ channel alpha subunit gene *KCNH3* (*BEC1* or *ELK2*) within our results, in line with our former rumination GWAS result *KCTD12* [11]. KCTD12 acts on the same pathway between GABA_B_ receptor and Kir3 K^+^ channel as G-protein subunit γ2 [81], implicated in our mapped genes. Former results were contradictory regarding folate dependence of prostaglandin D2 and E2 pathways [82,83,84] and of apolipoprotein B effects [85,86]; however, these pathways also emerged in our results in the optimal folate intake group.

Polygenic scores of rumination significantly explained some variance of the whole, validated rumination scale, but only if based on top risk variants within the optimal folate intake group. Although we have no information on folate intake or folate status in the target sample with the whole rumination scale, this result underlined the importance of differentiating between folate intake levels when considering genetics. Indeed, genetic background of rumination (and worry) in case of suboptimal folate intake explained less of rumination in the optimal folate intake subgroup than vice versa. This discrepancy may suggest that a high amount of genetic vulnerabilities may be compensated by an optimal folate intake. However, even the explained variance in the direction from optimal to suboptimal folate intake was around one tenth of SNP heritability, suggesting considerably different genetic underpinnings depending on folate intake level.

### 4.5. Limitations

Our study had some limitations. First of all, folate intake was assessed only once in 41.82%, and only as follow-up in 40.22% of all participants, with a 24 h dietary recall questionnaire with multiple pass method. However, the aim was to determine habitual dietary pattern, and validation studies suggest that this method is suitable to cost-effectively establish this information in large cohorts [87]. Second, a precise relative timing of dietary habits and perseverative cognition would be needed, serum and/or erythrocyte folate levels should be assessed, and further intervention studies should be implemented in order to validate the proposed mechanism of folate intake effects. Third, the lower sample size in the suboptimal folate intake group resulted in lower statistical power to determine genetic factors of rumination. However, the genetic pathways identified had well-known biological connections to folate metabolism, supporting their validity. Fourth, to validate our PRS results, a target sample with reliably measured folate intake and/or folate status will be required. Fifth, enrichment tests of mapped genes did not correct for LD between close genes. Therefore, our results with this methodology, though plausible, need future replication. 

## 5. Conclusions

Our results point to a more specific genetic background of past-focused perseverative negative thinking, in contrast to the more general “Are you a worrier?” item. Importance of the specificity of past-focus was also underlined by the shared genetics between this past-focused perseverative negative thinking item and the widely used RRS rumination scale across two different populations. Rumination can thus be regarded as a good candidate for a transdiagnostic endophenotype. Furthermore, rumination as a potential endophenotype could differentiate between different risk biological pathways in different folate intake groups. This is especially interesting because rumination did not show any association with folate intake itself, in contrast to the negative association between worry and folate intake. These results point to the importance of simultaneously taking into account genetic and environmental factors to determine personalized intervention in polygenic and multifactorial disorders. Inconsistent benefits of folate supplementation in depression [17,18,19,20], and a potential benefit of folate in anxiety disorders could also be reframed in light of our results.

## Figures and Tables

**Figure 1 nutrients-13-04396-f001:**
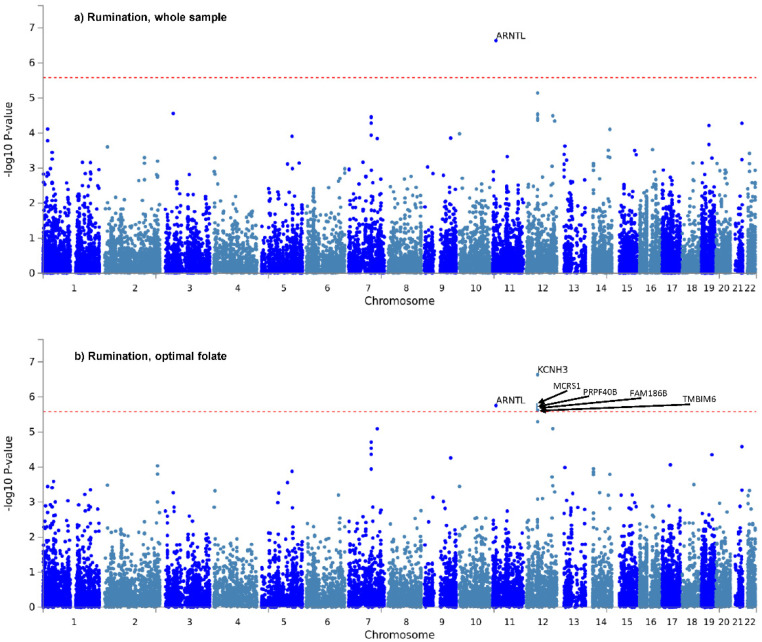
Manhattan plots of gene-based tests for “rumination” item in the whole study sample (**a**) and in the optimal folate intake group (**b**). −log10 of *p*-value is displayed in function of genomic position. Red line denotes significance threshold, corrected within only one analysis. In the whole sample (**a**) *ARNTL*, and in the optimal folate intake group (**b**) *KCNH3*, *MCRS1*, *ARNTL*, *PRPF40B*, *FAM186B* and *TMBIM6* survived this less strict correction.

**Figure 2 nutrients-13-04396-f002:**
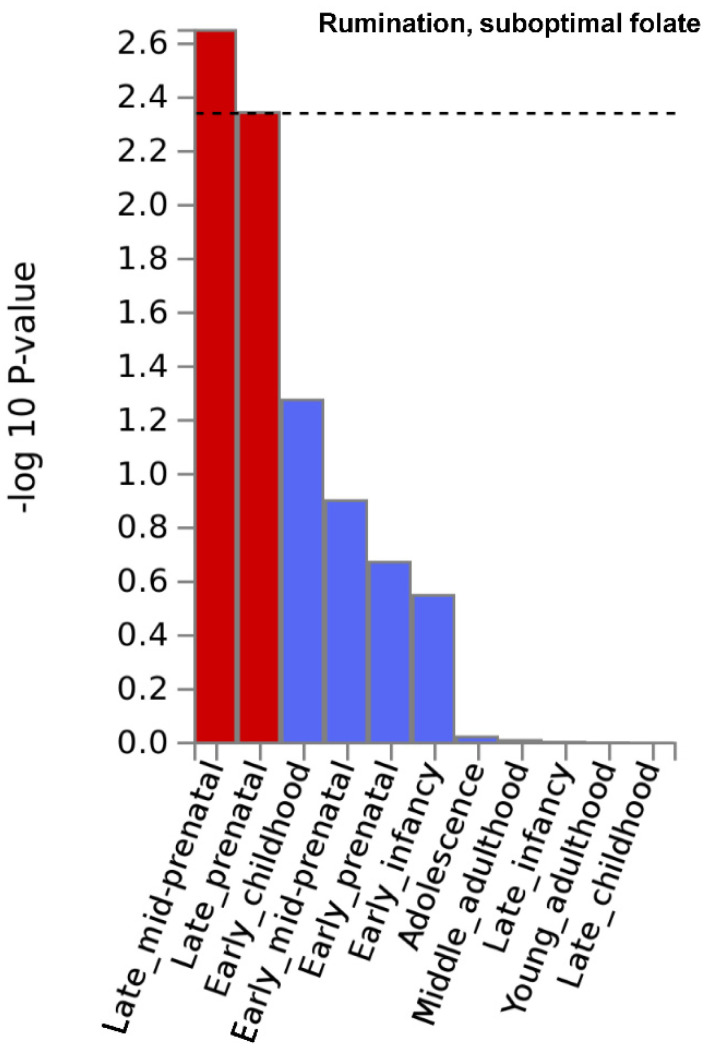
Brain developmental stage-specific expression of genes that are strongly associated with “rumination” in the suboptimal folate intake group. −log10 *p*-value is displayed for each of the 11 general developmental stages of BrainSpan’s brain development samples. Dashed line denotes significance threshold corrected for the 11 tissues only within this analysis. Results suggest that genes that showed strong association with rumination in the suboptimal folate group are highly expressed during late-mid and late prenatal brain development.

**Figure 3 nutrients-13-04396-f003:**
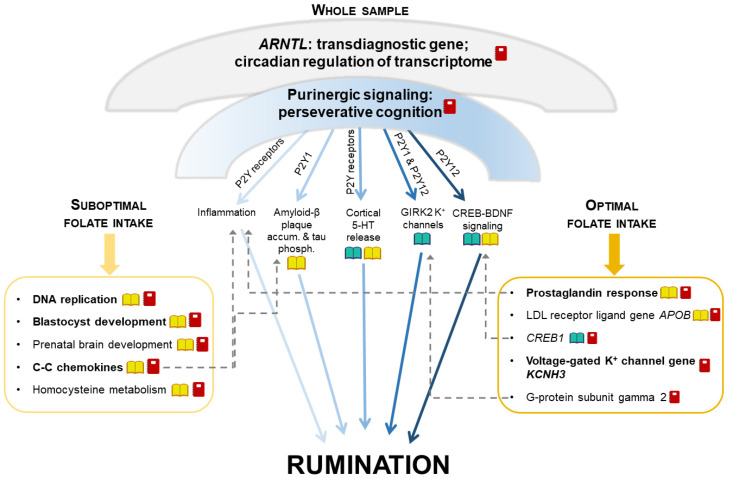
A proposed model of the impact of multifaceted mechanisms on rumination via additive effects of distinct narrower pathways. Umbrella-like biological mechanisms, being significant for rumination in our whole sample, can be decomposed into separate components that are affected by folate intake in a distinct extent. Umbrella-like mechanisms include *ARNTL* (that regulates more than 10% of the transcriptome in a circadian rhythm) and purinergic signaling. Separate components influenced by purinergic signaling are displayed below the umbrella, between the two boxes. The left box represents components significant in our suboptimal folate intake group, and the right box represents components significant in our optimal folate intake group. Some of the components represented in the boxes can be biologically related to the components influenced by purinergic signaling displayed between the two boxes (dashed arrows). Dark-light shading of arrows represents the extent to which each component influenced by purinergic signaling might be sensitive to folate intake. 
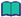
: biological pathways previously having been related to rumination; 
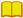
: biological pathways modulated by folate level; 
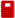
: our present study’s results for rumination (with bold font if surviving correction for multiple testing); GIRK2: G protein-activated inwardly rectifying potassium channel subunit 2; 5-HT: serotonin; CREB: cAMP-response element binding protein; BDNF: brain-derived neurotrophic factor; LDL: low density lipoprotein; accum.: accumulation; phosph.: phosphorylation.

**Table 1 nutrients-13-04396-t001:** Proportion of variance of “rumination” and “worry” items explained by all SNPs (SNP heritability) in the whole study sample, in the suboptimal folate intake group, and in the optimal folate intake group.

	“Do You Worry Too Long after an Embarrassing Experience?”		“Are You a Worrier?”	
	SNP h^2^	S.E. of SNP h^2^	SNP h^2^	S.E. of SNP h^2^
Whole sample	0.0286	0.0041	0.0339	0.0044
Suboptimal folate intake group	0.0433	0.0256	0.023	0.0246
Optimal folate intake group	0.0272	0.0046	0.0348	0.005

SNP: single-nucleotide polymorphism, h^2^: heritability, S.E.: standard error.

**Table 2 nutrients-13-04396-t002:** Most significant ten gene sets by MAGMA analysis, for “rumination” in the suboptimal folate intake group, and their ranks in the other five groups. GO_bp: biological processes, and GO_mf: molecular function subcategories of MsigDB C5 gene set collections.

for “Rumination”, Suboptimal Folate				Rank for “Rumination”		Rank for “Worry”		
Rank	Gene Set	Number of Genes	*p*-Value	Optimal Folate	Whole Sample	Suboptimal Folate	Optimal Folate	Whole Sample
1.	GO_mf:go_vascular_endothelial_growth_factor_binding	7	4.55 × 10^−6^	8786	3400	14,266	4007	4642
2.	Curated_gene_sets:petrova_prox1_targets_dn	56	4.17 × 10^−5^	9925	596	12,720	3257	8036
3.	GO_bp:go_response_to_nitric_oxide	19	4.91 × 10^−5^	209	137	1869	296	641
4.	GO_bp:go_diaphragm_development	9	1.64 × 10^−4^	12,105	7551	754	14,661	14,533
5.	GO_mf:go_platelet_derived_growth_factor_binding	11	5.28 × 10^−4^	412	100	3414	7910	1621
6.	GO_bp:go_positive_regulation_of_centriole_replication	6	5.28 × 10^−4^	6508	1879	8768	10,420	5176
7.	Curated_gene_sets:bandres_response_to_carmustin_without_mgmt_48hr_dn	30	6.51 × 10^−4^	12,804	9844	483	14,015	14,062
8.	Curated_gene_sets:zhan_multiple_myeloma_subgroups	31	8.33 × 10^−4^	6439	7505	9613	75	1219
9.	GO_bp:go_homocysteine_metabolic_process	12	8.48 × 10^−4^	5316	745	7168	10,786	13,211
10.	GO_bp:go_ductus_arteriosus_closure	5	9.93 × 10^−4^	12,033	6912	6821	9481	14,774

## Data Availability

The data presented in this study are available on request from the corresponding author. The data are not publicly available due to ethical considerations.

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
