# Peer review of "Biology of Perseverative Negative Thinking: The Role of Timing and Folate Intake"

_nutrients, 2021, doi:10.3390/nu13124396_

Round 1

Reviewer 1 Report

Eszlari and colleagues conducted a genome-wide association study employing data from the UK Biobank database. They showed how the
“rumination” and “worry” items of Eysenck Personality Inventory Neuroticism scale were associated with folate intake. 

The study is well conducted, with a large sample of data and adequate statistical procedures to assess significance. My suggestion is to reformulate in the Methods section using the formulation "FDR; p<0.05". I would also suggest authors to add details on the reasons related to their choices of those specific methods to correct for multiple comparisons and the threshold they applied. 

Further, I appreciated the brief discussion of the role of genes in brain development, while I also think that a further explanation of the role of the brain can be added. Indeed, I believe that the manuscript would benefit from a brief discussion of the adult and elder brains, for example referring to studies showing how hemodynamic activity can act as an intermediate phenotype in explaining the relationship between folate and vitamin b12 and cognition (e.g., https://doi.org/10.1155/2019/6874805). 

Also, in the abstract, the authors refer to the benefits of folate in anxiety disorders, while I think that this is not entirely highlighted in the author's discussion and conclusions. I would suggest expanding this aspect.

Lastly, I would also suggest authors to rephrase the limits section of their study, as there are different repetitions in the words used. 

Author Response

Response to Reviewer 1 Comments

Point 1/a: The study is well conducted, with a large sample of data and adequate statistical procedures to assess significance. My suggestion is to reformulate in the Methods section using the formulation "FDR; p<0.05".

Response 1/a: Thank you for your comments. We have reformulated all occurrences of FDR correction, in the manuscript file (see line 211 on page 5) as well as in Supplementary File 1 (three times on page 4 and one time on page 17, in “All Markup” view).

Point 1/b: I would also suggest authors to add details on the reasons related to their choices of those specific methods to correct for multiple comparisons and the threshold they applied.

Response 1/b: We used the built-in Benjamini-Hochberg FDR correction method in “GENE2FUNC” analysis of FUMA, and the built-in permutation method in PRSice-2. In other analyses, where built-in corrections for multiple testing were not available, we used the strictest Bonferroni method, correcting for not only the number of SNPs, genes, gene sets or tissues, but for the number of tests (two outcomes by three groups) as well (as detailed in the Analyses section of Supplementary File 1, on pages 3-4 in “All Markup” view). We added these reasons in 2.4. Analyses subsection of Materials and Methods section through lines 189-217 (pages 5-6) of the manuscript.

Point 2: Further, I appreciated the brief discussion of the role of genes in brain development, while I also think that a further explanation of the role of the brain can be added. Indeed, I believe that the manuscript would benefit from a brief discussion of the adult and elder brains, for example referring to studies showing how hemodynamic activity can act as an intermediate phenotype in explaining the relationship between folate and vitamin b12 and cognition (e.g., https://doi.org/10.1155/2019/6874805).

Response 2: We added a paragraph on this suggested topic in the 4.3. Genetic determinants of past-focused perseverative negative thinking in case of suboptimal folate intake subsection of Discussion (lines 459-470 on page 13).

Point 3: Also, in the abstract, the authors refer to the benefits of folate in anxiety disorders, while I think that this is not entirely highlighted in the author's discussion and conclusions. I would suggest expanding this aspect.

Response 3: Thank you, we have added a paragraph on this topic in the 4.1. Time perspective and event-specificity of perseverative negative thinking subsection of Discussion (lines 371-387 on page 11).

Point 4: Lastly, I would also suggest authors to rephrase the limits section of their study, as there are different repetitions in the words used.

Response 4: We have rephrased Limitations to avoid repetitions (lines 500-513 on page 14).

Reviewer 2 Report

The manuscript (nutrients-1452043) entitled “Biology of perseverative negative thinking: the role of timing and folate intake” is by Nora Eszlari, et al. The authors claimed to investigate genomic background of two aspects (Past-oriented rumination and future-oriented worry) of perseverative negative thinking within separate groups of individuals with suboptimal versus optimal folate intake. They claimed that there are possible benefits of folate in anxiety disorders, and argued to simultaneously take into account genetic and environmental factors to determine personalized intervention in polygenic and multifactorial disorders.

Comments:

  1. Please describe your reasons to include only “72 621 white British participants from UK Biobank”. Were they the only samples available?
  2. What were the rationales to include or exclude certain populations?
  3. What were the folate intake levels in other populations? Or other disease groups? Was there a difference?
  4. What was the reason showing table 2? Have the authors analyzed what the contributions of those pathways in the preservative negative thinking?
  5. The information shown in Figure 3 does not explain how suboptimal folate intake leads to those consequences. From that point of view, the value of Figure 3 is limited.
  6. An intervention experiment should be done to prove the findings.

Author Response

Response to Reviewer 2 Comments

Point 1: Please describe your reasons to include only “72 621 white British participants from UK Biobank”. Were they the only samples available? 

Response 1: We have rephrased inclusion criteria in lines 105-118 (pages 3-4) of the manuscript, so that it is more straightforward now that the main limitation of our sample size was the subset of participants completing the Oxford WEbQ dietary questionnaire, then other limitations were to pass genomic quality control and then to have non-missing data on all relevant variables.

Point 2: What were the rationales to include or exclude certain populations?

Response 2: To avoid false positive genotype-phenotype associations emerging because of different allele frequencies in different geographical (as well as cultural) regions, populations of distinct genetic ancestry should not be mixed in genetic analyses. As we have now indicated in lines 112-113 of page 4, we chose the White British population because it constitutes the vast majority of the UK Biobank database. Each of the other populations is represented within UK Biobank with a very limited sample size (see https://biobank.ndph.ox.ac.uk/showcase/field.cgi?id=21000), and having sample sizes large enough is very crucial in genetic / genomic analyses to avoid type-II errors. Moreover, UK Biobank has genetic ancestry data only on Caucasians (see https://biobank.ndph.ox.ac.uk/showcase/field.cgi?id=22006).

Point 3: What were the folate intake levels in other populations? Or other disease groups? Was there a difference?

Response 3: We cannot give a proper answer to the second question because our analyses focused not on disease groups but on phenotypes that can confer a risk for a variety of diseases. However, we can answer to the first and third parts of the question. A total of 7133 subjects were filtered out from our initial, Oxford WEbQ-based database, because of the filtering steps detailed in lines 107-110 of page 3. Comparing frequencies of suboptimal and optimal folate intake between the two groups of participants included and not included in our present analyses, we get a non-significant (p=0.675) Fisher’s exact test, suggesting no difference in folate intake according to inclusion in our present analyses. Counts, expected counts and percentages are provided here:

not present in final dataset

present in final dataset

Total

Folate intake

below 200 µg/day

Count

1058

10638

11696

Expected Count

1046.1

10649.9

11696.0

% within column

14.8%

14.6%

14.7%

above 200 µg/day

Count

6075

61983

68058

Expected Count

6086.9

61971.1

68058.0

% within column

85.2%

85.4%

85.3%

Total

Count

7133

72621

79754

Expected Count

7133.0

72621.0

79754.0

% within column

100.0%

100.0%

100.0%

Point 4: What was the reason showing table 2? Have the authors analyzed what the contributions of those pathways in the preservative negative thinking?

Response 4: Now we have added some explanation on the link between folate and homocysteine metabolism in line 265 of page 7. We included Table 2 in the manuscript because we considered it important to show that homocysteine pathway (as aggregated from gene-based results, as detailed in lines 194-199 of page 5) achieves a high significance only in the suboptimal folate intake group and only for “rumination” as outcome.

Point 5: The information shown in Figure 3 does not explain how suboptimal folate intake leads to those consequences. From that point of view, the value of Figure 3 is limited.

Response 5: Thank you for this comment. Indeed, Figure 3 is a visual representation of our present knowledge regarding different pathomechanisms that might be associated with rumination based on genetic association results – ours presented in this manuscript and previously published findings. Left side of the figure represents how suboptimal folate intake might contribute to these complex processes, although to draw causal inferences further experimental investigations are required, as the reviewer pointed out below. Now we have added some clarifications in lines 395-399 of page 11, on the role of former studies, present results and the needed future studies in Figure 3.

Point 6: An intervention experiment should be done to prove the findings.

Response 6: Now we have added this note in lines 502-503 of page 14, in the Limitations section.

Round 2

Reviewer 2 Report

I have no more comments.